# Consistency and Monotonicity Regularization for Neural Knowledge Tracing

## Abstract

Knowledge Tracing (KT), tracking a human's knowledge acquisition, is a central component in online learning and AI in Education. In this paper, we present a simple, yet effective strategy to improve the generalization ability of KT models: we propose three types of novel data augmentation, coined replacement, insertion, and deletion, along with corresponding regularization losses that impose certain consistency or monotonicity biases on model's predictions for the original and augmented sequence. Extensive experiments on various KT benchmarks show that our regularization scheme consistently improves the model performances, under 3 widely-used neural networks and 4 public benchmarks, e.g., it yields 6.3% improvement in AUC under the DKT model and the ASSISTmentsChall dataset.

## 1 Introduction

In recent years, Artificial Intelligence in Education (AIEd) has gained much attention as one of the currently emerging fields in educational technology. In particular, the recent COVID-19 pandemic has transformed the setting of education from classroom learning to online learning. As a result, AIEd has become more prominent because of its ability to diagnose students automatically and provide personalized learning paths. High-quality diagnosis and educational content recommendation require good understanding of students' current knowledge status, and it is essential to model their learning behavior precisely. Due to this, Knowledge Tracing (KT), a task of modeling a student's evolution of knowledge over time, has become one of the most central tasks in AIEd research.

Since the work of Piech et al. (2015), deep neural networks have been widely used for the KT modeling. Current research trends in the KT literature concentrate on building more sophisticated, complex and large-scale models, inspired by model architectures from Natural Language Processing (NLP), such as LSTM (Hochreiter & Schmidhuber, 1997) or Transformer (Vaswani et al., 2017) architectures, along with additional components that extract question textual information or students' forgetting behaviors (Huang et al., 2019; Pu et al., 2020; Ghosh et al., 2020). However, as the number of parameters of these models increases, they may easily overfit on small datasets and hurt model's generalizabiliy. Such an issue has been under-explored in the literature.

To address the issue, we propose simple, yet effective data augmentation strategies for improving the generalization ability of KT models, along with novel regularization losses for each strategy. In particular, we suggest three types of data augmentation, coined (skill-based) replacement, insertion, and deletion. Specifically, we generate augmented (training) samples by randomly replacing questions that a student solved with similar questions or inserting/deleting interactions with fixed responses. Then, during training, we impose certain consistency (for replacement) and monotonicity (for insertion/deletion) bias on a model's predictions by optimizing corresponding regularization losses that compares the original and the augmented interaction sequences. Here, our intuition behind the proposed consistency regularization is that the model's output for two interaction sequences with same response logs for similar questions should be close. Next, the proposed monotonicity regularization is designed to enforce the model's prediction to be monotone with respect to the number of questions that correctly (or incorrectly) answered, i.e., a student is more likely to answer correctly (or incorrectly) if the student did the same more in the past. By analyzing distribution of the previous correctness rates of interaction sequences, we can observe that the existing student interaction datasets indeed have monotonicity properties - see Figure 1 and Section A.2 for details. The overall augmentation and regularization strategies are sketched in Figure 2. Such regularization strategies

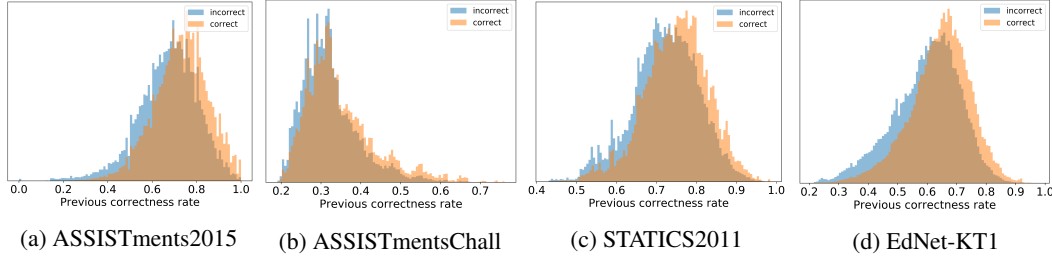

(a) ASSISTments2015  (b) ASSISTmentsChall  (c) STATICS2011  (d) EdNet-KT1

Figure 1: Distribution of the correctness rate of past interactions when the response correctness of current interaction is fixed, for 4 knowledge tracing benchmark datasets. Orange (resp. blue) represents the distribution of correctness rate (of past interactions) where current interaction's response is correct (resp. incorrect). $x$ axis represents previous interactions' correctness rates (values in $[0, 1]$). The orange distribution lean more to the right than the blue distribution, which shows the monotonicity nature of the interaction datasets. See Section A.2 for details.

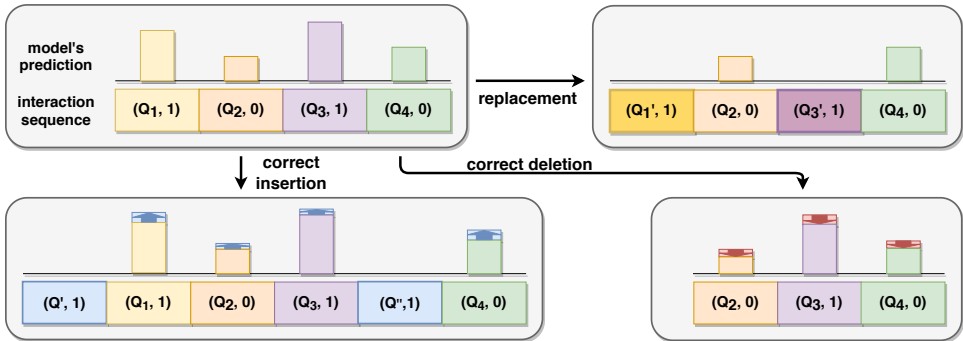

Figure 2: Augmentation strategies and corresponding bias on model's predictions (predicted correctness probabilities). Each tuple represents question id and response of the student's interaction (1 means correct). Replacing interactions with similar questions ($Q_1, Q_3$ to $Q'_1, Q'_3$) does not change model's predictions drastically. Introducing new interactions with correct responses ($Q', Q''$) increases model's estimation , but deleting such interaction ($Q_1, 1$) decreases model's estimation.

are motivated from our observation that existing knowledge tracing models' prediction often fails to satisfy the consistency and monotonicity condition, e.g., see Figure 4 in Section 3.

We demonstrate the effectiveness of the proposed method with 3 widely used neural knowledge tracing models - DKT (Piech et al., 2015), DKVMN (Zhang et al., 2017b), and SAINT (Choi et al., 2020a) - on 4 public benchmark datasets - ASSISTments2015, ASSISTmentsChall, STATICS2011, and EdNet-KT1. Extensive experiments show that, regardless of dataset or model architecture, our scheme remarkably increases the prediction performance - 6.2% gain in Area Under Curve (AUC) for DKT on the ASSISTmentsChall dataset. In particular, ours is much more effective under smaller datasets: by using only 25% of the ASSISTmentsChall dataset, we improve AUC of the DKT model from 69.68% to 75.44%, which even surpasses the baseline performance 74.4% with the full training set. We further provide various ablation studies for the selected design choices, e.g., AUC of the DKT model on the ASSISTments2015 dataset is dropped from 72.44% to 66.48% when we impose 'reversed' (wrong) monotonicity regularization. We believe that our work can be a strong guideline for other researchers attempting to improve the generalization ability of KT models.

## 1.1 RELATED WORKS AND PRELIMINARIES

**Data augmentation** is arguably the most trustworthy technique to prevent overfitting or improve the generalizability of machine learning models. In particular, it has been developed as an effective way to impose a domain-specific, inductive bias to a model. For example, for computer vision models, simple image warpings such as flip, rotation, distortion, color shifting, blur, and random erasing are the most popular data augmentation methods (Shorten & Khoshgoftaar, 2019). More advanced techniques, e.g., augmenting images by interpolation (Zhang et al., 2017a; Yun et al., 2019) or by using generative adversarial networks (Huang et al., 2018), have been also investigated. For NLP models,

it is popular to augment texts by replacing words with synonyms (Zhang et al., 2015) or words with similar (contextualized) embeddings (Wang & Yang, 2015; Kobayashi, 2018). As an alternative method, back translation (Sennrich et al., 2016; Yu et al., 2018) generates an augmented sentence by translating a given sentence into a different language domain and translate it back to the original domain with machine translation models. Recently, Wei & Zou (2019) show that even simple methods like random insertion/swap/deletion could improve text classification performances. In the area of speech recognition, vocal tract length normalization (Jaitly & Hinton, 2013), synthesizing noisy audio (Hannun et al., 2014), perturbing speed (Ko et al., 2015), and augmenting spectrogram (Park et al., 2019) are popular as data augmentation methods.

The aforementioned data augmentation techniques have been used not only for standard supervised learning setups, but also for various unsupervised and semi-supervised learning frameworks, by imposing certain inductive biases to models. For example, consistency learning (Sajjadi et al., 2016; Xie et al., 2019; Berthelot et al., 2019; Sohn et al., 2020) impose a consistency bias to a model so that the model's output is invariant under the augmentations, by means of training the model with consistency regularization loss (e.g. $L^2$-loss between outputs). Abu-Mostafa (1992; 1990) suggested general theory for imposing such inductive biases (which are stated as *hints*) via additional regularization losses. Their successes highlight the importance of domain specific knowledge for designing appropriate data augmentation strategies, but such results are rare in the domain of AIEd, especially for Knowledge Tracing.

**Knowledge tracing** (KT) is the task of modeling student knowledge over time based on the student's learning history. Formally, for a given student interaction sequence $(I_1, \ldots, I_T)$, where each $I_t = (Q_t, R_t)$ is a pair of question id $Q_t$ and the student's response correctness $R_t \in \{0, 1\}$ (1 means correct), KT aims to estimate the following probability

$$\mathbb{P}[R_t = 1 | I_1, I_2, \ldots, I_{t-1}, Q_t], \tag{1}$$

i.e., the probability that the student answers correctly to the question $Q_t$ at $t$-th step. Corbett & Anderson (1994) proposed Bayesian Knowledge Tracing (BKT) that models a student's knowledge as a latent variable in a Hidden Markov Model. Also, various seq2seq architectures including LSTM (Hochreiter & Schmidhuber, 1997), MANN (Graves et al., 2016), and Transformer (Vaswani et al., 2017) are used in the context of KT and showed their efficacy. Deep Knowledge Tracing (DKT) is the first deep learning based model that models student's knowledge states as LSTM's hidden state vectors (Piech et al., 2015). Dynamic Key-Value Memory Network and its variation can exploit relationships between questions/skills with concept vectors and concept-state vectors with key and value matrices, which is more interpretable than DKT (Zhang et al., 2017b; Abdelrahman & Wang, 2019). Transformer based models (Pandey & Karypis, 2019; Choi et al., 2020a; Ghosh et al., 2020; Pu et al., 2020) are able to learn long-range dependencies with their self-attention mechanisms and be trained in parallel. Utilizing additional features of interactions, such as question texts (Huang et al., 2019; Pandey & Srivastava, 2020), prerequisite relations (Chen et al., 2018) and time information (Nagatani et al., 2019; Choi et al., 2020a; Pu et al., 2020) is another way to improve performances. Recent works try to use graph neural networks (Nakagawa et al., 2019; Liu et al.; Tong et al., 2020; Yang et al., 2020b) and convolutional networks (Yang et al., 2020a; Shen et al., 2020) to model relations between questions and skills or extract individualized prior knowledge.

## 2 CONSISTENCY AND MONOTONICITY REGULARIZATION FOR KT

For a given set of data augmentations $\mathcal{A}$, we train KT models with the following loss:

$$\mathcal{L}_{\text{tot}} = \mathcal{L}_{\text{ori}} + \sum_{\text{aug} \in \mathcal{A}} (\lambda_{\text{aug}} \mathcal{L}_{\text{aug}} + \lambda_{\text{reg-aug}} \mathcal{L}_{\text{reg-aug}}), \tag{2}$$

where $\mathcal{L}_{\text{ori}}$ is the commonly used binary cross-entropy (BCE) loss for original training sequences and $\mathcal{L}_{\text{aug}}$ are the same BCE losses for augmented sequences generated by applying augmentation strategies $\text{aug} \in \mathcal{A}$.[1] $\mathcal{L}_{\text{reg-aug}}$ are the regularization losses that impose consistency and monotonicity bias on model's predictions for the original and augmented sequence, which are going to be defined in the following sections. Finally, $\lambda_{\text{aug}}, \lambda_{\text{reg-aug}} > 0$ are hyperparameters to control the trade-off among $\mathcal{L}_{\text{ori}}, \mathcal{L}_{\text{aug}}$, and $\mathcal{L}_{\text{reg-aug}}$.

---

[1]For replacement and insertion, we do not include outputs for augmented interactions in $\mathcal{L}_{\text{aug}}$.

In the following sections, we introduce our three simple augmentation strategies, replacement, insertion and deletion with corresponding consistency and monotonicity regularization losses, $\mathcal{L}_{\text{reg-rep}}$, $\mathcal{L}_{\text{reg-cor\_ins}}$ (or $\mathcal{L}_{\text{reg-incor\_ins}}$) and $\mathcal{L}_{\text{reg-cor\_del}}$ (or $\mathcal{L}_{\text{reg-incor\_del}}$), respectively.

## 2.1 REPLACEMENT

Replacement, which is motivated by the synonym replacement in NLP, is an augmentation strategy that replaces questions in the original interaction sequence with other similar questions *without changing their responses*, where *similar questions* are defined as the questions that have overlapping skills attached to. Our hypothesis is that the predicted correctness probabilities for questions in an augmented interaction sequence will not change a lot from those in the original interaction sequence. Formally, for each interaction in the original interaction sequence $(I_1, \ldots, I_T)$, we randomly decide whether the interaction will be replaced or not, following the Bernoulli distribution with the probability $\alpha_{\text{rep}}$. If an interaction $I_t = (Q_t, R_t)$ with a set of skills $S_t$ associated with the question $Q_t$ is set to be replaced, we determine $I_t^{\text{rep}} = (Q_t^{\text{rep}}, R_t)$ by selecting a question $Q_t^{\text{rep}}$ with its associated set of skills $S_t^{\text{rep}}$ that satisfies $S_t \cap S_t^{\text{rep}} \neq \emptyset$. The resulting augmented sequence $(I_1^{\text{rep}}, \ldots, I_T^{\text{rep}})$ is generated by replacing $I_t$ with $I_t^{\text{rep}}$ for $t \in \mathbf{R} \subset [T] = \{1, 2, \ldots, T\}$, where $\mathbf{R}$ is a set of indices to replace. Then we consider the following consistency regularization loss:

$$\mathcal{L}_{\text{reg-rep}} = \mathbb{E}_{t \notin \mathbf{R}}[(p_t - p_t^{\text{rep}})^2] \tag{3}$$

where $p_t$ and $p_t^{\text{rep}}$ are model's predicted correctness probabilities for $t$-th question of the original and augmented sequences, respectively. We do not include the output for the replaced interactions in the loss computation. For the replacement strategy itself we consider several variants. For instance, randomly selecting a question for $Q_t^{\text{rep}}$ from the question pool is an alternative strategy if a skill set for each question is not available. It is also possible to only consider outputs for interactions that are replaced or consider outputs for all interactions in the augmented sequence for the loss computation. We investigate the effectiveness of each strategy in Section 3.

## 2.2 INSERTION

When a student answers more questions correctly (resp. incorrectly), the predicted correctness probabilities of the KT models for the remaining questions should increase (resp. decrease). Based on this intuition, we introduce a *monotonicity constraint* by inserting new interactions into the original interaction sequence. Formally, we generate an augmented interaction sequence $(I_1^{\text{ins}}, \ldots, I_{T'}^{\text{ins}})$ by inserting a correctly (resp. incorrectly) answered interaction $I_t^{\text{ins}} = (Q_t^{\text{ins}}, 1)$ (resp. $I_t^{\text{ins}} = (Q_t^{\text{ins}}, 0)$) into the original interaction sequence $(I_1, \ldots, I_T)$ for $t \in \mathbf{I} \subset [T']$, where the question $Q_t^{\text{ins}}$ is randomly selected from the question pool and $\mathbf{I}$ with the size $\alpha_{\text{ins}}$ proportion of the original sequence is a set of indices of inserted interactions. Then our hypothesis is formulated as $p_t \leq p_{\sigma(t)}^{\text{ins}}$ (resp. $p_t \geq p_{\sigma(t)}^{\text{ins}}$), where $p_t$ and $p_t^{\text{ins}}$ are model's predicted correctness probabilities for $t$-th question of the original and augmented sequences, respectively. Here, $\sigma : [T] \rightarrow [T'] - \mathbf{I}$ is the order-preserving bijection which satisfies $I_t = I_{\sigma(t)}^{\text{ins}}$ for $1 \leq t \leq T$. (For instance, in Figure 2, $\sigma$ sends $\{1, 2, 3, 4\}$ to $\{2, 3, 4, 6\}$) We impose our hypothesis through the following losses:

$$\mathcal{L}_{\text{reg-cor\_ins}} = \mathbb{E}_{t \in [T]}[\max(0, p_t - p_{\sigma(t)}^{\text{ins}})], \quad \mathcal{L}_{\text{reg-incor\_ins}} = \mathbb{E}_{t \in [T]}[\max(0, p_{\sigma(t)}^{\text{ins}} - p_t)] \tag{4}$$

where $\mathcal{L}_{\text{reg-cor\_ins}}$ and $\mathcal{L}_{\text{reg-incor\_ins}}$ are losses for augmented interaction sequences of inserting correctly and incorrectly answered interactions, respectively.

## 2.3 DELETION

Similar to the insertion augmentation strategy, we propose another monotonicity constraint by removing some interactions in the original interaction sequence based on the following hypothesis: if a student's response records contain less correct (resp. incorrect) answers, the correctness probabilities for the remaining questions would become decrease (resp. increase). Formally, from the original interaction sequence $(I_1, \ldots, I_T)$, we randomly sample a set of indices $\mathbf{D} \subset [T]$, where $R_t = 1$ (resp. $R_t = 0$) for $t \in \mathbf{D}$, based on the Bernoulli distribution with the probability $\alpha_{\text{del}}$. We remove the index $t \in \mathbf{D}$ and impose the hypothesis $p_t \geq p_{\sigma(t)}^{\text{del}}$ (resp. $p_t \leq p_{\sigma(t)}^{\text{del}}$), where $p_t$ and $p_t^{\text{del}}$ are model's predicted correctness probabilities for $t$-th question of the original and augmented

| dataset | model | no augmentation | insertion + deletion | insertion + deletion + replacement |
|---|---|---|---|---|
| ASSIST2015 | DKT | $72.01 \pm 0.05$ | $\mathbf{72.46} \pm 0.06$ | $72.39 \pm 0.07$ |
| | DKVMN | $71.21 \pm 0.09$ | $72.00 \pm 0.18$ | $\mathbf{72.23} \pm 0.09$ |
| | SAINT | $72.13 \pm 0.09$ | $72.78 \pm 0.06$ | $\mathbf{72.81} \pm 0.04$ |
| ASSISTChall | DKT | $74.40 \pm 0.16$ | $75.98 \pm 0.07$ | $\mathbf{79.07} \pm 0.08$ |
| | DKVMN | $74.46 \pm 0.11$ | $75.06 \pm 0.10$ | $\mathbf{78.21} \pm 0.05$ |
| | SAINT | $77.01 \pm 0.18$ | $78.02 \pm 0.09$ | $\mathbf{80.18} \pm 0.05$ |
| STATICS2011 | DKT | $86.43 \pm 0.29$ | $87.18 \pm 0.12$ | $\mathbf{87.27} \pm 0.11$ |
| | DKVMN | $84.89 \pm 0.17$ | $85.65 \pm 0.94$ | $\mathbf{87.17} \pm 0.14$ |
| | SAINT | $85.82 \pm 0.50$ | $86.53 \pm 0.30$ | $\mathbf{87.56} \pm 0.06$ |
| EdNet-KT1 | DKT | $72.75 \pm 0.09$ | $74.04 \pm 0.04$ | $\mathbf{74.28} \pm 0.06$ |
| | DKVMN | $73.58 \pm 0.08$ | $73.94 \pm 0.05$ | $\mathbf{74.16} \pm 0.11$ |
| | SAINT | $74.78 \pm 0.05$ | $\mathbf{75.32} \pm 0.05$ | $75.26 \pm 0.02$ |

Table 1: Performances (AUCs) of DKT, DKVMN, and SAINT models on 4 public benchmark datasets. The results show the mean and standard deviation averaged over 5 runs and the best result for each dataset and model is indicated in bold.

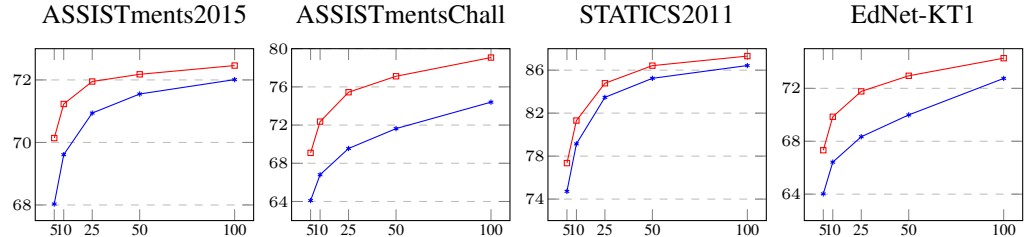

Figure 3: Performances with various sizes of training data under the DKT model. $x$ axis stands for the portion of the training set we use for training (relative to the full train set) and $y$ axis is the AUC. Blue line represents the AUCs of the vanilla DKT model, and red line represents the AUCs of the DKT model trained with augmentations and regularizations.

sequences, respectively. Here, $\sigma : [T] - \mathbf{D} \to [T']$ is the order preserving bijection with $I_t = I_{\sigma(t)}^{\mathrm{del}}$ for $t \in [T] - \mathbf{D}$. We impose the hypothesis through the following losses:

$$\mathcal{L}_{\mathrm{reg\text{-}cor\_del}} = \mathbb{E}_{t \notin \mathbf{D}}[\max(0, p_{\sigma(t)}^{\mathrm{del}} - p_t)], \quad \mathcal{L}_{\mathrm{reg\text{-}incor\_del}} = \mathbb{E}_{t \notin \mathbf{D}}[\max(0, p_t - p_{\sigma(t)}^{\mathrm{del}})] \quad (5)$$

where $\mathcal{L}_{\mathrm{reg\text{-}cor\_del}}$ and $\mathcal{L}_{\mathrm{reg\text{-}incor\_del}}$ are losses for augmented interaction sequences of deleting correctly and incorrectly answered interactions, respectively.

## 3 EXPERIMENTS

We demonstrated the effectiveness of the proposed method on 4 widely used benchmark datasets: ASSISTments2015, ASSISTmentsChall, STATICS2011, and EdNet-KT1. ASSISTments datasets are the most widely used benchmark for Knowledge Tracing, which is provided by ASSISTments online tutoring platform[2] (Feng et al., 2009). There are several versions of dataset depend on the years they collected, and we used ASSISTments2015[3] and ASSISTmentsChall[4]. ASSISTmentsChall dataset is provided by the 2017 ASSISTments data mining competition. STATICS2011 consists of the interaction logs from an engineering statics course, which is available on the PSLC datashop[5]. EdNet-KT1 is the largest publicly available interaction dataset consists of TOEIC (Test of English for Interational Communication) problem solving logs collected by *Santa*[6] (Choi et al.,

---

[2]https://new.assistments.org/

[3]https://sites.google.com/site/assistmentsdata/home/2015-assistments-skill-builder-data

[4]https://sites.google.com/view/assistmentsdatamining

[5]https://pslcdatashop.web.cmu.edu/DatasetInfo?datasetId=507

[6]https://aitutorsanta.com/

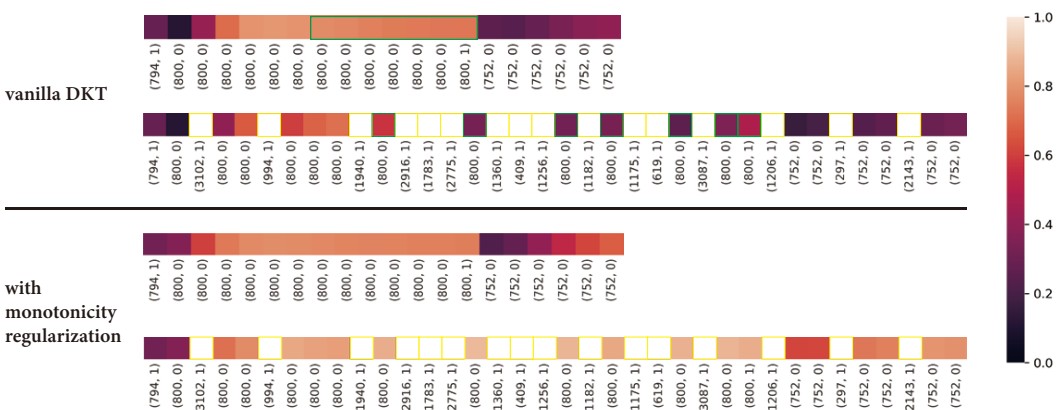

Figure 4: Response correctness prediction for a student in the ASSISTmentsChall dataset. We randomly insert interactions with correct responses (interactions with yellow boundaries). In case of the vanilla DKT model, the predictions for the original interactions (especially the interactions with green boundaries) are decreased, even if the student answered more questions correctly. However, such problem is resolved when we train the model with monotonicity regularization (with the loss $\mathcal{L}_{\text{tot}} = \mathcal{L}_{\text{ori}} + \mathcal{L}_{\text{cor\_ins}} + 100 \cdot \mathcal{L}_{\text{reg-cor\_ins}}$). Unlike the vanilla DKT model, predicted correctness probabilities for the original interactions are increased after insertion.

2020b). We reduce the size of the EdNet-KT1 dataset by sampling 6000 users among 600K users. Detailed statistics and pre-processing methods for these datasets are described in Appendix. With the exception of the EdNet-KT1 dataset, we used 80% of the students as a training set and the remaining 20% as a test set.

We test DKT (Piech et al., 2015), DKVMN (Zhang et al., 2017b), and SAINT (Choi et al., 2020a) models. For DKT, we set the embedding dimension and the hidden dimension as 256. For DKVMN, key, value, and summary dimension are all set to be 256, and we set the number of latent concepts as 64. SAINT has 2 layers with hidden dimension 256, 8 heads, and feed-forward dimension 1024. All the models do not use any additional features of interactions except question ids and responses as an input, and the model weights are initialized with Xavier distribution (Glorot & Bengio, 2010). They are trained from scratch with batch size 64, and we use the Adam optimizer (Kingma & Ba, 2015) with learning rate 0.001 which is scheduled by Noam scheme with warm-up step 4000 as Vaswani et al. (2017) suggest. We set each model's maximum sequence size as 100 on ASSISTments2015 & EdNet-KT1 dataset and 200 on ASSISTmentsChall & STATICS2011 dataset. Hyperparameters for augmentations, $\alpha_{\text{aug}}$, $\lambda_{\text{reg-aug}}$, and $\lambda_{\text{aug}}$ are searched over $\alpha_{\text{aug}} \in \{0.1, 0.3, 0.5\}$, $\lambda_{\text{reg-aug}} \in \{1, 10, 50, 100\}$, and $\lambda_{\text{aug}} \in \{0, 1\}$. For all dataset, we evaluate our results using 5-fold cross validation and use Area Under Curve (AUC) as an evaluation metric.

## 3.1 MAIN RESULTS

The results (AUCs) are shown in Table 1 that compares models without and with augmentations, and we report the best results for each strategy. (The detailed hyperparameters for these results are given in Appendix.) The 4th column represents results using both insertion and deletion, and the last column shows the results with all 3 augmentations. Since there's no big difference on performance gain between insertion and deletion, we only report the performance that uses one or both of them together. We use skill-based replacement if skill information for each question in the dataset is available, and use question-random replacement that that selects new questions among all questios if not (e.g. ASSISTments2015). As one can see, the models trained with consistency and monotonicity regularizations outperforms the models without augmentations in a large margin, regardless of model's architectures or datasets. Using all three augmentations gives the best performances for most of the cases. For instance, there exists 6.3% gain in AUC on ASSISTmentsChall dataset under the DKT model. Furthermore, not only enhancing the prediction performances, our training scheme also resolves the vanilla model's issue where the monotonicity condition on the predictions of original and augmented sequences is violated. As in Figure 4, the predictions of the model trained with monotonicity regularization (correct insertion) are increased after insertion, which contrasts to the vanilla DKT model's outputs.

| dataset | loss | replacement | correct insertion | incorrect insertion | correct deletion | incorrect deletion |
|---|---|---|---|---|---|---|
| ASSIST2015 | (6) | $72.03 \pm 0.06$ | $71.98 \pm 0.06$ | $71.98 \pm 0.05$ | $72.05 \pm 0.04$ | $72.04 \pm 0.02$ |
| $(72.01 \pm 0.05)$ | (2) | $72.06 \pm 0.03$ | $72.09 \pm 0.06$ | $72.35 \pm 0.11$ | $72.53 \pm 0.08$ | $72.26 \pm 0.04$ |
| ASSISTChall | (6) | $75.13 \pm 0.04$ | $74.61 \pm 0.17$ | $74.57 \pm 0.14$ | $74.92 \pm 0.12$ | $74.42 \pm 0.20$ |
| $(74.40 \pm 0.16)$ | (2) | $78.85 \pm 0.08$ | $75.98 \pm 0.07$ | $75.64 \pm 0.12$ | $75.60 \pm 0.06$ | $74.77 \pm 0.11$ |
| STATICS2011 | (6) | $86.89 \pm 0.23$ | $86.45 \pm 0.26$ | $86.40 \pm 0.22$ | $86.53 \pm 0.29$ | $86.55 \pm 0.25$ |
| $(86.43 \pm 0.29)$ | (2) | $87.27 \pm 0.11$ | $86.72 \pm 0.23$ | $87.18 \pm 0.12$ | $87.07 \pm 0.33$ | $86.97 \pm 0.26$ |
| EdNet-KT1 | (6) | $73.04 \pm 0.10$ | $72.81 \pm 0.08$ | $72.88 \pm 0.09$ | $72.99 \pm 0.07$ | $73.28 \pm 0.04$ |
| $(72.75 \pm 0.09)$ | (2) | $73.89 \pm 0.06$ | $73.73 \pm 0.06$ | $73.52 \pm 0.06$ | $74.04 \pm 0.04$ | $73.76 \pm 0.04$ |

Table 2: Comparison of the performances (AUCs) of the DKT model, trained with only data augmentation (i.e., using the loss (6)) and with consistency and monotonicity regularizations (i.e., using the loss (2)). AUCs of the vanilla DKT model are given in parentheses below the dataset names.

| augmentation | direction | ASSIST2015 | ASSISTChall | STATICS2011 | EdNet-KT1 |
|---|---|---|---|---|---|
| - | - | $72.01 \pm 0.05$ | $74.40 \pm 0.16$ | $86.43 \pm 0.29$ | $72.75 \pm 0.09$ |
| correct insertion | increase | $72.08 \pm 0.02$ | $75.98 \pm 0.06$ | $86.72 \pm 0.23$ | $73.70 \pm 0.08$ |
| incorrect insertion | decrease | $72.31 \pm 0.04$ | $75.34 \pm 0.16$ | $87.18 \pm 0.12$ | $73.40 \pm 0.06$ |
| correct deletion | decrease | $72.44 \pm 0.05$ | $75.60 \pm 0.06$ | $87.07 \pm 0.33$ | $74.01 \pm 0.05$ |
| incorrect deletion | increase | $72.26 \pm 0.04$ | $74.77 \pm 0.11$ | $86.68 \pm 0.27$ | $73.71 \pm 0.04$ |
| correct insertion | decrease | $71.79 \pm 0.06$ | $75.42 \pm 0.17$ | $86.58 \pm 0.50$ | $69.67 \pm 0.06$ |
| incorrect insertion | increase | $70.73 \pm 0.10$ | $74.92 \pm 0.11$ | $86.22 \pm 0.18$ | $71.95 \pm 0.15$ |
| correct deletion | increase | $66.48 \pm 0.10$ | $74.68 \pm 0.13$ | $86.76 \pm 0.27$ | $71.23 \pm 0.81$ |
| incorrect deletion | decrease | $67.34 \pm 0.17$ | $73.91 \pm 0.14$ | $86.58 \pm 0.28$ | $69.99 \pm 0.11$ |

Table 3: Ablation test on the directions of monotonicity regularizations with the DKT model. 2nd to 5th rows show the results with the original regularization losses, and the last 4 rows show the results with the reversed regularization losses.

Since overfitting is expected to be more severe when using a smaller dataset, we conduct experiments using various fractions of the existing training datasets (5%, 10%, 25%, 50%) and show that our augmentations yield more significant improvements for smaller training datasets. Figure 3 shows performances of DKT model on various datasets, with and without augmentations. For example, on ASSISTmentsChall dataset, using 100% of the training data gives AUC 74.4%, while the same model trained with augmentations achieved AUC 75.44% with only 25% of the training dataset.

## 3.2 ABLATION STUDY

**Are constraint losses necessary?** One might think that data augmentations are enough for boosting up the performance, and imposing consistency and monotonicity are not necessary. However, we found that including such regularization losses for training is essential for further performance gain. To see this, we compare the performances of the model trained only with KT losses for both original and augmented sequences

$$\mathcal{L}_{\text{tot}} = \mathcal{L}_{\text{ori}} + \sum_{\text{aug} \in \mathcal{A}} \lambda_{\text{aug}} \mathcal{L}_{\text{aug}} \qquad (6)$$

(where $\lambda_{\text{aug}} = 1$) and with consistency and monotonicity regularization losses (2) where $\mathcal{A}$ is a set that contains a single augmentation. Training a model with the loss (6) can be thought as using augmentations without imposing any consistency or monotonicity biases.

Table 2 shows results under the DKT model. Using only data augmentation (training the model with the loss (6)) gives a marginal gain in performances or even worse performances. However, training with both data augmentation and consistency or monotonicity regularization losses (2) give significantly higher performance gain. Under ASSISTmentsChall dataset, using replacement along with

| dataset | no augmentation | replaced inters | remaining inters | full inters | qDKT |
|---------|-----------------|-----------------|------------------|-------------|------|
| ASSIST2015 | $72.01 \pm 0.05$ | $70.53 \pm 0.07$ | $\mathbf{72.07} \pm 0.02$ | $71.39 \pm 0.09$ | - |
| ASSISTChall | $74.40 \pm 0.16$ | $74.68 \pm 0.09$ | $\mathbf{78.45} \pm 0.08$ | $75.91 \pm 0.07$ | $75.17 \pm 0.13$ |
| STATICS2011 | $86.43 \pm 0.29$ | $82.97 \pm 0.27$ | $\mathbf{87.17} \pm 0.15$ | $83.49 \pm 0.10$ | $86.51 \pm 0.10$ |
| EdNet-KT1 | $72.75 \pm 0.09$ | $65.52 \pm 0.07$ | $\mathbf{73.87} \pm 0.10$ | $68.77 \pm 0.11$ | $64.49 \pm 0.09$ |

Table 4: Performances (AUCs) of the DKT model with variations of replacements and qDKT with Lapacian regularization. Best result for each dataset is indicated in bold.

| dataset | no augmentation | question-random | interaction-random | skill-set | skill |
|---------|-----------------|-----------------|---------------------|-----------|-------|
| ASSIST2015 | $72.01 \pm 0.05$ | $\mathbf{72.07} \pm 0.02$ | $71.77 \pm 0.05$ | - | - |
| ASSISTChall | $74.40 \pm 0.16$ | $78.39 \pm 0.09$ | $74.27 \pm 0.38$ | $77.57 \pm 0.08$ | $\mathbf{78.45} \pm 0.08$ |
| STATICS2011 | $86.43 \pm 0.29$ | $86.35 \pm 0.06$ | $84.50 \pm 0.28$ | - | $\mathbf{87.17} \pm 0.15$ |
| EdNet-KT1 | $72.75 \pm 0.09$ | $73.84 \pm 0.05$ | $72.62 \pm 0.17$ | $73.80 \pm 0.07$ | $\mathbf{73.87} \pm 0.10$ |

Table 5: Performances (AUCs) of the DKT model with different type of replacements - question-random replacement, interaction-random replacement, skill-set-based replacement, and skill-based replacement. Best result for each dataset is indicated in bold.

consistency regularization improves AUC by 6%, which is much higher than the 1% improvement only using data augmentation.

**Ablation on monotonicity constraints** We perform an ablation study to compare the effects of monotonicity regularization and *reversed* monotonocity regularization. Monotonocity regularization introduces constraint loss to align the inserted or deleted sequence in order to modify the probability of correctness of the original sequence to follow insertion or deletion. For example, when a correct response is inserted to the sequence, the probability of correctness for the original sequence increases. Reversed monotonocity regularization modifies the probability of correctness in the opposite manner, where inserting a correct response would decrease the probability of correctness in the original sequence.

For each aug $\in \{\mathrm{cor\_ins}, \mathrm{incor\_ins}, \mathrm{cor\_del}, \mathrm{incor\_del}\}$, we can define reversed version of the monotonicity regularization loss $\mathcal{L}_{\mathrm{reg\text{-}aug}}^{\mathrm{rev}}$ which impose the opposite constraint on the model's output, e.g. we define $\mathcal{L}_{\mathrm{reg\text{-}cor\_ins}}^{\mathrm{rev}}$ as

$$\mathcal{L}_{\mathrm{reg\text{-}cor\_ins}}^{\mathrm{rev}} = \mathbb{E}_{t \in [T]}[\max(0, p_{\sigma(t)}^{\mathrm{ins}} - p_t)] = \mathcal{L}_{\mathrm{reg\text{-}incor\_ins}} \tag{7}$$

which forces model's output of correctness probability to *decrease* when *correct* responses are inserted. In this experiments, we do not include KT loss for augmented sequences (set $\lambda_{\mathrm{aug}} = 0$) to observe the effects of consistency loss only. Also, the same hyperparameters ($\alpha_{\mathrm{aug}}$ and $\lambda_{\mathrm{reg\text{-}aug}}$) are used for both the original and reversed constraints.

Table 3 shows the performances of DKT model with the original and reversed monotonicity regularizations. Second row represents the performance with no augmentations, the 3rd to the 6th rows represent the results from using original (aligned) insertion/deletion monotonicity regularization losses, and the last four rows represent the results when the reversed monotonicity regularization losses are used. The results demonstrate that using aligned monotonicity regularization loss outperforms the model with reversed monotonicity regularization loss. Also, the performances of reversed monotonicity shows large decrease in performance on several datasets even compared to the model with no augmentation. For example, in case of the EdNet-KT1 dataset, the model's performance with correct insertion along with original (aligned) regularization improves the AUC from 72.75% to 73.70%, while using the reversed regularization drops the performance to 69.67%.

**Ablation on replacement.** We compare our consistency regularization with the other two variations of replacements - consistency regularization on replaced interactions and full interactions - and qDKT (Sonkar et al., 2020). As we mentioned in Section 3, there are two more possible variations of the consistency loss for the replacement depends on whether we include replaced interaction's output in the loss or not:

$$\mathcal{L}_{\mathrm{reg\text{-}rep\_ro}} = \mathbb{E}_{t \in \mathbf{R}}[(p_t - p_t^{\mathrm{rep}})^2], \quad \mathcal{L}_{\mathrm{reg\text{-}rep\_full}} = \mathbb{E}_{t \in [T]}[(p_t - p_t^{\mathrm{rep}})^2], \tag{8}$$

where ro stands for *replaced only*. We compared such variations with the original consistency loss $\mathcal{L}_{\mathrm{reg\text{-}rep}}$ that does not include predictions for the replaced interactions. For all variations, we used

the same replacement probability $\alpha_{\text{rep}}$ and loss weight $\lambda_{\text{reg-rep}}$, and we do not include KT loss for replaced sequences (set $\lambda_{\text{rep}} = 0$) as before. Also, we compare replacement with qDKT that uses the following Laplacian loss which regularizes the variance of predicted correctness probabilities for questions that fall under the same skill:

$$\mathcal{L}_{\text{Laplacian}} = \mathbb{E}_{(q_i, q_j) \in \mathcal{Q} \times \mathcal{Q}}[\mathbf{1}(i, j)(p_i - p_j)^2] \tag{9}$$

where $\mathcal{Q}$ is the set of all questions, $p_i, p_j$ are the model's predicted correctness probabilities for the questions $q_i, q_j$, and $\mathbf{1}(i, j)$ is 1 if $q_i, q_j$ have common skills attached, otherwise 0. It is similar to our variation of consistency regularization ($\mathcal{L}_{\text{reg-rep\_ro}}$ in (8)) that only compares replaced interactions' outputs, but it does not replace questions and it compares all questions (with same skills) at once. Since the hyperparameter $\lambda$ that scales the Laplacian regularization term is not provided in the paper, we use the same set of hyperparameters we use for other losses, and report the best results among them. Table 4 shows that including the replaced interactions' outputs hurt performances. For example, under the EdNet-KT1 dataset, all the variations of consistency regularization and Laplacian regularization significantly dropped AUCs to under 70%, while the original consistency regularization boost up the performance from 72.75% to 73.87%.

To see the effect of using the skill information of questions for replacement, we compared skill-based replacement with three different random versions of replacement: *question random replacement*, *interaction random replacement*, and *skill-set-based replacement*. For *question random replacement*, we replace questions with different ones randomly (without considering skill information), while *interaction random replacement* changes both question and responses (sample each response with 0.5 probability). *Skill-set-based replacement* is almost the same as the original skill-based replacement, but the candidates of the questions to be replaced are chosen as ones with exactly same set of skills are associated, not only have common skills ($S = S^{\text{rep}}$). The results in Table 5 show that the performances of the question random replacements depends on the nature of dataset. It shows similar performance with skill-based replacement on ASSISTmentsChall and EdNet-KT1 datasets, but only give a minor gain or even dropped the performance on other datasets. However, applying interaction-random replacement significantly hurt performances over all datasets, e.g. the AUC is decreased from 86.43% to 84.50% on STATICS2011 dataset. This demonstrates the importance of fixing responses of the interactions for consistency regularization. At last, skill-set-based replacement works similar or even worse than the original skill-based replacement. Note that each question of the STATICS2011 dataset has single skill attached to, so the performance of skill-based and skill-set-based replacement coincide on the dataset.

## 4 CONCLUSION

We propose simple augmentation strategies with corresponding constraint regularization losses for KT and show their efficacy. We only considered the most basic features of interactions, question and response correctness, and other features like elapsed time or question texts enables us to exploit diverse augmentation strategies if available. Furthermore, exploring applicability of our idea on other AiEd tasks (dropout prediction or at-risk student prediction) is another interesting future direction.

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

# A  APPENDIX

## A.1  DATASET STATISTICS AND PRE-PROCESSING

Detailted dataset statistics are given in the Table 6.

- **ASSISTments**: For ASSISTments2015 dataset, we filtered out the logs with CORRECTS not in $\{0, 1\}$. Note that ASSISTments2015 dataset only provides question and no corresponding skills.

- **STATICS2011** A concatenation of a problem name and step name is used as a question id, and the values in the column KC (F2011) are regarded as skills attached to each question.

- **EdNet-KT1** Among 600K students, we filtered out whose interaction length is in $[100, 1000]$, and randomly sampled 6000 users, where 5000 users for training and 1000 users for test.

| name | logs | students | questions | skills | avg. length | avg. correctness |
|------|------|----------|-----------|--------|-------------|------------------|
| ASSIST2015 | 683801 | 19840 | 100 | - | 34.47 | 0.73 |
| ASSISTChall | 942816 | 1709 | 3162 | 102 | 551.68 | 0.37 |
| STATICS2011 | 261937 | 333 | 1224 | 81 | 786.60 | 0.72 |
| EdNet-KT1 | 2051701 | 6000 | 14419 | 188 | 341.95 | 0.63 |

Table 6: Dataset statistics.

## A.2  MONOTONICITY NATURE OF DATASETS

We perform data analysis to explore monotonicity nature of datasets, i.e. a property that students are more likely to answer correctly if they did the same more in the past. For each interaction of each student, we see the distribution of past interactions' correctness rate. Formally, for given interaction sequences $(I_1, \ldots, I_T)$ with $I_t = (Q_t, R_t)$ and each $2 \leq t \leq T$, we compare the distributions of past interactions' correctness rate

$$\text{correctness\_rate}_{<t} = \frac{1}{t-1} \sum_{\tau=1}^{t-1} \mathbf{1}_{R_\tau = 1}$$

where $\mathbf{1}_{R_\tau = 1}$ is an indicator function which is 1 (resp. 0) when $R_\tau = 1$ (resp. $R_\tau = 0$). We compare the distributions of correctness\_rate$_{<t}$ over all interactions with $R_t = 1$ and $R_t = 0$ separately, and the results are shown in Figure 1. We can see that the distributions of previous correctness rates of interactions with correct response lean more to the right than ones of interactions with incorrect response. This shows the positive correlation between previous correctness rate and the current response correctness, and it also explains why monotonicity regularization actually improve prediction performances of knowledge tracing models.

## A.3  MODEL'S PREDICTIONS AND CONSISTENCY REGULARIZATION LOSSES

Instead of analyzing consistency nature of datasets directly, we compare the test consistency loss for correctly and incorrectly predicted responses separately, with the DKT model on ASSISTmentsChall, STATICS2011, and EdNet-KT1 datasets. Table 7 shows the average consistency loss for correctly and incorrectly predicted responses, with the vanilla DKT model and the model trained with consistency regularization losses. When we compute the test consistency loss, we replaced each (previous) interaction's questions to another questions with overlapping skills with $\alpha_{\text{rep}} = 0.3$ probability. For all models, the average loss for the correctly predicted responses are lower than the incorrectly predicted responses. This verifies that smaller consistency loss actually improves prediction accuracy.

| dataset | target response | vanilla | regularized |
|---------|-----------------|---------|-------------|
| ASSISTChall | correct | 0.01028 | 0.00027 |
|  | incorrect | 0.01713 | 0.00039 |
| STATICS2011 | correct | 0.00618 | 0.00049 |
|  | incorrect | 0.01748 | 0.00093 |
| EdNet-KT1 | correct | 0.00422 | 0.00091 |
|  | incorrect | 0.00535 | 0.00116 |

Table 7: Comparison of the average consistency loss for correctly and incorrectly predicted responses of the DKT model.

## A.4 HYPERPARAMETERS

Table 8 describes detailed hyperparameters for each augmentation and model that are used for the main results (Table 1). Each entry represents a tuple of augmentation probability ($\alpha_{\text{aug}}$) and a weight for constraint loss ($\lambda_{\text{reg-aug}}$), which shows the best performances among $\alpha_{\text{aug}} \in \{0.1, 0.3, 0.5\}$ and $\lambda_{\text{reg-aug}} \in \{1, 10, 50, 100\}$. Each entry represents ($\alpha_{\text{aug}}, \lambda_{\text{reg-aug}}$) for each augmentation. We use $\lambda_{\text{aug}} = 1$ for all experiments with augmentations, except for the DKT model on STATICS2011 dataset with incorrect insertion augmentation ($\lambda_{\text{incor\_ins}} = 0$).

To see the effect of augmentation probabilities and regularization loss weights, we perform grid search over $\alpha_{\text{aug}} \in \{0.1, 0.3, 0.5\}$ and $\lambda_{\text{reg-aug}} \in \{1, 10, 50, 100\}$ with DKT model, and the AUC results are shown as heatmaps in Figure 5.

| dataset | model | insertion + deletion | | | | insertion + deletion + replacement | | | | |
|---------|-------|---------|----------|---------|----------|---------|----------|---------|----------|------|
|  |  | cor_ins | incor_ins | cor_del | incor_del | cor_ins | incor_ins | cor_del | incor_del | rep |
| ASSIST2015 | DKT | (0, 0) | (0, 0) | (0.3, 100) | (0, 0) | (0.3, 100) | (0, 0) | (0, 0) | (0, 0) | (0.1, 10) |
|  | DKVMN | (0.5, 100) | (0, 0) | (0, 0) | (0, 0) | (0.5, 100) | (0, 0) | (0, 0) | (0, 0) | (0.3, 1) |
|  | SAINT | (0, 0) | (0.5, 10) | (0, 0) | (0, 0) | (0, 0) | (0.5, 10) | (0, 0) | (0, 0) | (0.3, 1) |
| ASSISTChall | DKT | (0.5, 100) | (0, 0) | (0, 0) | (0, 0) | (0.5, 1) | (0, 0) | (0, 0) | (0, 0) | (0.3, 100) |
|  | DKVMN | (0.5, 1) | (0, 0) | (0, 0) | (0, 0) | (0.5, 1) | (0, 0) | (0, 0) | (0, 0) | (0.5, 100) |
|  | SAINT | (0, 0) | (0, 0) | (0.3, 1) | (0, 0) | (0, 0) | (0.3, 1) | (0.3, 1) | (0, 0) | (0.3, 100) |
| STATICS2011 | DKT | (0, 0) | (0.5, 10) | (0, 0) | (0, 0) | (0, 0) | (0, 0) | (0, 0) | (0, 0) | (0.3, 100) |
|  | DKVMN | (0, 0) | (0, 0) | (0.3, 10) | (0, 0) | (0, 0) | (0, 0) | (0.3, 1) | (0, 0) | (0.3, 10) |
|  | SAINT | (0, 0) | (0.5, 1) | (0, 0) | (0.5, 1) | (0, 0) | (0.5, 1) | (0, 0) | (0.5, 1) | (0.3, 100) |
| EdNet-KT1 | DKT | (0, 0) | (0, 0) | (0.3, 50) | (0, 0) | (0, 0) | (0.3, 1) | (0.3, 1) | (0, 0) | (0.1, 100) |
|  | DKVMN | (0, 0) | (0.5, 1) | (0, 0) | (0, 0) | (0, 0) | (0.5, 1) | (0, 0) | (0, 0) | (0.1, 1) |
|  | SAINT | (0, 0) | (0.3, 50) | (0, 0) | (0, 0) | (0, 0) | (0.3, 50) | (0, 0) | (0, 0) | (0.5, 1) |

Table 8: Hyperparameters for Table 1.

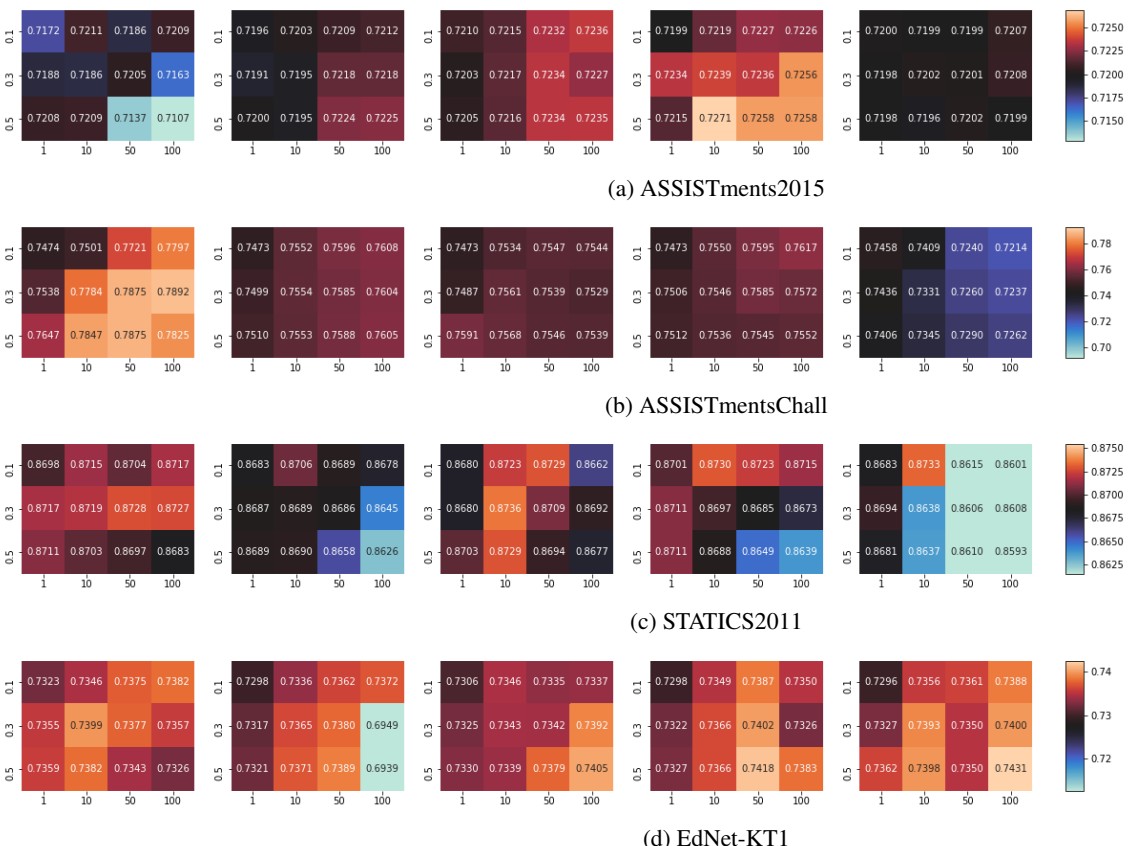

Figure 5: Performances (AUCs) of the DKT model for each augmentation and corresponding regularization with different augmentation probabilities ($\alpha_{\mathrm{aug}}$)) and regularization loss weights ($\lambda_{\mathrm{reg-aug}}$). The hyperparameters are searched over $\alpha_{\mathrm{aug}} \in \{0.1, 0.3, 0.5\}$ and $\lambda_{\mathrm{reg-aug}} \in \{1, 10, 50, 100\}$. For each dataset, each column represents results with replacement, correct insertion, incorrect insertion, correct deletion, and incorrect deletion, from left to right. We set $\lambda_{\mathrm{aug}} = 1$ for all cases. We use question-random replacement for ASSISTments2015 dataset.

