# OpenReview forum: "Consistency and Monotonicity Regularization for Neural Knowledge Tracing"
_ICLR.cc/2021/Conference — Reject_

### Official Review · AnonReviewer1 · 2020-10-27
**Not significant improvement over baseline**

**Rating:** 4
**Confidence:** 3

**Review:**

Disclaimer: I am not familiar with the educational AI field. With strong argumentation, or if one of my co-reviewers is an expert in the field, I could be persuaded to change my score.

The paper investigates how well data augmentation can help improve the performance of contemporary deep learning models for Knowledge Tracing, which is a key task for educational AI. The authors propose three different types of data augmentation strategies: replacement, insertion, and deletion. The authors provide a detailed experimental section with ablation studies, highlighting the benefits of using their model in addition to recently proposed models. The authors provide confidence intervals for their results proving the significance of their solution.

What this paper excels at is the breath and scale of their experimental section. I like that they have taken a large set of different datasets, ablated all improvements, and tried different training set size models.

However, I am have the following concerns that leads me to reject this paper:

- SAINT seems to be the most modern model, which is why this is particularly interesting as I assume any improvements would indicate a SOTA in the field. Though, in table 1, for the EdNet-KT1 dataset the authors report SAINT to have 74.78, while the original SAINT paper reports 78.11 AUC

- I do not see any arguments for why these augmentation methods are of specific interest. What motivated you to try this? beyond just wanting to add noise in the training. I believe adding some noise could give a small improvement, but I do not believe that such finding on a niche NLP subfield is of general interest to the scientific community beyond a workshop.

- In general, it is my understanding that this is 3 augmentation functions, something similar to dropout or synonym replacement. I think the explanation of these methods are overly complicated. I would like to see the authors making their method section easier to read and reduce the amount of unnecessary notation.

And some more specific comments:

- "However, as the number of parameters of these models increases, e.g., the recent GPT-3 model (Brown et al., 2020) has 175 billion parameters, they may easily overfit on small datasets and hurt model’s generalizabiliy. Such an issue has been under-explored in the literature" - The GPT-3 model doesnt overfit. Also I don't think massive language models are relevant to your problem. If it's the overfitting issue I would find something that reports on overfitting and the use of data augmentation to remedy it.

- I don't get Figure 1.

- What is consistency and contrastive learning? you reference 7 papers, but give no intuition about it's relevance to your work. Please elaborate.

- I dont get figure 2 when reading the paper from end-to-end, I don't think it should be on the top of page 3 when it's referenced in the results section.

- The math in 2.1 is unclear to me.

- What is the metric in table 1? ACC or AUC? there's a huge difference. Also, is it on the validation or test set?

- Are the ablation studies on the validation or test set?

UPDATE:

Thank you for clarifying the ethics concern. However, this makes it much more difficult to assess whether I believe your method works as well as you state. After having read the rebuttal and the other reviews, I am more confident that the methodology proposed lacks connection to educational relevance and novelty for publication at this venue. My score stays the same.

---

> ### Author Response · Authors · 2020-11-12
> **Response to R1 (1/2)**
>
> Thank you for your valuable efforts and time spent on reading our paper. Our responses to all your questions and comments are provided below.
>
> Q1. SAINT seems to be the most modern model, which is why this is particularly interesting as I assume any improvements would indicate a SOTA in the field. Though, in table 1, for the EdNet-KT1 dataset the authors report SAINT to have 74.78, while the original SAINT paper reports 78.11 AUC
>
> A1. The gap between two AUC results is due to the size of the EdNet-KT1 dataset used in the experiments. We reduce the size of EdNet-KT1 dataset in our experiments (see Appendix 1 for the details) because the full data is extremely large and often took more than 2 weeks to train in our machine. To the best of our knowledge, all prior works [1, 2] using EdNet-KT1 also reduced the dataset as like ours by randomly sampling the students, except for those by the original SAINT’ authors, due to the issue. As we propose a regularization scheme for knowledge tracing models, the regime of smaller datasets is more valuable and our major interest to test. It also allows more comprehensive ablation analysis to perform. Nevertheless, we will report the results for the full EdNet-KT1 dataset as much as possible in the revision and the final draft.
>
> Q2. I do not see any arguments for why these augmentation methods are of specific interest. What motivated you to try this? beyond just wanting to add noise in the training. I believe adding some noise could give a small improvement, but I do not believe that such finding on a niche NLP subfield is of general interest to the scientific community beyond a workshop. In general, it is my understanding that this is 3 augmentation functions, something similar to dropout or synonym replacement. I think the explanation of these methods are overly complicated. I would like to see the authors making their method section easier to read and reduce the amount of unnecessary notation.
>
> A2. We first emphasize that our focus is to design a new regularization method specialized to knowledge tracing (KT). To this end, we consider data augmentation methods (with appropriate losses) as they are popular ways to impose domain-specific bias, i.e., consistency or monotonicity bias for KT. This is impossible by using some generic, domain-agnostic regularization methods such as dropout or synonym replacement, and they are not that effective for KT. We provide more details in what follows.
>
> The replacement, insertion, and deletion augmentations are sufficiently different from some generic noisy training methods in NLP, i.e., as R2 mentioned, there’s something more that we can say about these augmentations in the context of KT. Skill-based replacements and fixed-response insertion & deletion, along with consistency and monotonicity biases, reflects the nature of the student interaction sequences. For example, deletion is not exactly the same as the dropout - we “compare” the model’s predictions of the original and deleted interaction sequence and impose monotonicity bias on it via monotonicity loss we defined (equation 5). The ablation studies on the monotonicity constraints in Section 3.2 shows that the direction of the monotonicity bias is important, which is also the unique nature of KT.
>
> Following your suggestion, we will simplify the explanations of our method in the revision.
>
> Q3. "However, as the number of parameters of these models increases, e.g., the recent GPT-3 model (Brown et al., 2020) has 175 billion parameters, they may easily overfit on small datasets and hurt model’s generalizabiliy. Such an issue has been under-explored in the literature" - The GPT-3 model doesnt overfit. Also I don't think massive language models are relevant to your problem. If it's the overfitting issue I would find something that reports on overfitting and the use of data augmentation to remedy it.
>
> A3. We agree that the sentence can be misleading for some readers and we will revise it. We meant that existing knowledge tracing (KT) models are variants of NLP models (e.g., RNN, Transformers), but KT models are typically trained under a much smaller dataset. Hence, it may cause an overfitting issue. Figure 3 supports that overfitting indeed occurs for KT models, as our method is more effective for smaller datasets in overall.
>
> Q4. I don’t get Figure 1.
>
> A5. We will update Figure 1 in the revision. The heights of each bar represents the model's predicted correctness probability, and the upper-left are the predictions for the original interaction sequence. The remaining three bar graphs represent models’ predictions for the augmented interactions sequences, which has consistency and monotonicity biases. For example, introducing new interactions with correct responses would increase the model’s predicted correctness probability for the remaining questions, which is a bias that can be achieved by training the model with our monotonicity losses.

---

> > ### Author Response · Authors · 2020-11-12
> > **Response to R1 (2/2)**
> >
> > Q5. What is consistency and contrastive learning? You reference 7 papers, but give no intuition about its relevance to your work. Please elaborate.
> >
> > A5. We mention consistency and contrastive learning in the related work section as data augmentation is also a key component for them. Their successes highlight the importance of domain-specific data augmentation, which motivates our work to design such augmentations for knowledge tracing (KT). Nevertheless, we will clarify this and add more explanation for consistency and contrastive learning in the revision.
> >
> > Q6. I don’t get figure 2 when reading the paper from end-to-end, I don’t think it should be on the top of page 4 when it’s referenced in the results section.
> >
> > A6. Thank you for your suggestion. The figure should be on the above of the results’ section as you mentioned. We will move it as you suggested in the revision.
> >
> > Q7. The math in 2.1 is unclear to me.
> >
> > A7. We will clarify the math in the revision. It would be very much appreciated and helpful for us if you could give some advice on which math/notation is unclear. Thank you!
> >
> > Q8. What is the metric in table 1? ACC or AUC? There’s a huge difference. Also, it is on the validation or test set?
> >
> > A8. All the experiments results given in the papers are AUCs, which is the most common evaluation metric for the knowledge tracing, on a test set. For each benchmark datasets, we randomly choose 20% of whole students as a test set, and evaluate our results using 5-fold cross validation. We will add ‘AUC’ to all captions of figures in the revision.
> >
> > Q9. Are the ablation studies on the validation or test set?
> >
> > A9. The AUC results of ablations studies are on the test set.
> >
> >
> > [1] Liu et. al. Improving Knowledge Tracing via Pre-training Question Embeddings, IJCAI 2020.
> > [2] Yang et. al. GIKT: A Graph-based Interaction Model for Knowledge Tracing, arXiv preprint.

---

### Official Review · AnonReviewer2 · 2020-10-28
**Good experimental results, sensible technique, some additional citations suggested**

**Rating:** 7
**Confidence:** 4

**Review:**

This paper presents some enhancements for Knowledge Tracing (KT), in which predictions are made about the odds of a student answering a question correctly given a sequence of correct/incorrect responses to previous questions.  The authors observe that the predictive model should obey certain 3 common sense constraints. If a question is replaced in the student's data by a very similar question, the prediction should not change much. If an additional correct question is added to the data, the odds of the student being correct on the next question should go up, and the odds should go down for questions being removed and/or added with incorrect responses.  The learning algorithm's objective function is augmented with additional terms which encourage the model to obey these constraints.

Pros:

The method for the most part makes sense. The experiments are reasonably thorough (4 benchmark datasets are tested) and non-trivial accuracy gains are demonstrated, although dramatic gains are only achieved on 1 of the 4 benchmarks.  Ablation experiments provide additional confidence that the interpretation of the impact of the method is correct.

Cons:

The paper should cite previous work from the 1990s from Yaser Abu-Mostafa, who pioneered the use of these kinds of 'regularization' enhancements under the name of 'hints'. See e.g. ' A Method for Learning from Hints', NeurIPs 1993, and several similar papers.  A Google Scholar search of other ML work on monotonicity may also be beneficial if the authors seek to continue this line of research. See e.g. the recent work of Maya Gupta et. al.

 I would have appreciated more information about the 'skill sets' associated with each question and how that impacts the replacement.  The authors say question is chosen as a replacement if it has some skill overlap with the original question (page 4).  However, if there are multiple skills associated with the question, wouldn't it make more sense to choose replacements based on percentage skill overlap than a simple binary detection of any overlap?

Further comments:

One comment I have (and I recognize that not everyone)

Some typos:

Impose certain consistency or monotonicity bias on model’s predictions -> biases on the model’s predictions

Fig 2 randomly insert intractions – interactions…

even the student answered more questions correctly -> even if the student answered more questions correctly

when other researchers will pursue to improve the generalization ability of KT models in the future-> for other researchers attempting to improve the generalization ability of KT models.

Section 3.1 among all questios -> among all questions

---

> ### Author Response · Authors · 2020-11-12
> **Response to R2**
>
> Thank you for your valuable efforts and time spent on reading our paper. Our responses to all your questions and comments are provided below.
>
> Q1. The paper should cite previous work from the 1990s from Yaser Abu-Mostafa, who pioneered the use of these kinds of 'regularization' enhancements under the name of 'hints'. See e.g. ' A Method for Learning from Hints', NeurIPs 1993, and several similar papers. A Google Scholar search of other ML work on monotonicity may also be beneficial if the authors seek to continue this line of research. See e.g. the recent work of Maya Gupta et. al.
>
> A1. Thank you for letting us know about the works. We will add these references to the revision.
>
> Q2. I would have appreciated more information about the 'skill sets' associated with each question and how that impacts the replacement. The authors say question is chosen as a replacement if it has some skill overlap with the original question (page 4). However, if there are multiple skills associated with the question, wouldn't it make more sense to choose replacements based on percentage skill overlap than a simple binary detection of any overlap?
>
> A2. Thank you for your suggestion. In the case of ASSISTments2015 and STATICS2011, unique skills are associated with each question, but the questions of ASSISTmentsChall and EdNet-KT1 have multiple skills attached. In the revision, we are going to do further experiments on replacement that considers multiple skills as you suggested.

---

> > ### Comment · AnonReviewer2 · 2020-11-24
> > **Thanks for adding the references**
> >
> > Much appreciated !

---

### Official Review · AnonReviewer3 · 2020-10-29

**Rating:** 6
**Confidence:** 2

**Review:**

The authors show that various forms of augmentations can improve the performance on knowledge tracing. The experiments are conducted on ASSIST2015, ASSISTChall, STATICS2011 and EdNet-KT1. Data augmentation leads to a certain amount of improvements. However, consistency training provides more significant improvements.

The novelty part is limited since the proposed methods such as insertion, deletion and replacements are intuitive and also seen in prior works in NLP. The monotonicity constraint is specific to the knowledge tracing task though.

The improvements are consistent, and especially significant when the training data is limited. More ablation studies on the hyperparameters would be beneficial.

My major concern is that the novelty is limited. The paper tackles a less well-studied task so more experiments should be added. For example,
1. Would consistency training leads to more improvements when the training data is limited?
2. How would the hyperparameter in insertion, deletion and replacements impact the performance?
3. Would more advanced augmentation lead to better performance?

---

> ### Author Response · Authors · 2020-11-12
> **Response to R3**
>
> Thank you for your valuable efforts and time spent on reading our paper. Our responses to all your questions and comments are provided below.
>
> Q1. The novelty part is limited since the proposed methods such as insertion, deletion and replacements are intuitive and also seen in prior works in NLP. The monotonicity constraint is specific to the knowledge tracing task though.
>
> A1. We view the simplicity of our method as strength instead of weakness. Furthermore, our replacement, insertion, and deletion augmentations are sufficiently different from those in NLP and cannot be directly applied to the NLP tasks. We provide more details in what follows.
>
> In the context of knowledge tracing (KT), similar questions, i.e. questions with same skill attached, is an analogue of synonyms in NLP. However, student interaction consists of two components: question “and” response correctness. Hence replacing interactions rather than questions is more difficult and nontrivial than replacing word tokens. Also, existing works on consistency regularization on NLP tasks usually replace whole sentences with similar sentences [1, 2], where we replace only a few of the given interaction sequence, and compare the predictions of non-replaced interactions. The ablation studies on replacement in Section 3.2 (Table 4 and Table 5) shows that our proposed setup - applying consistency loss only for the non-replaced tokens (Equation 3) and only replacing questions but not responses - significantly outperformed its variations.
>
> As you mentioned, monotonicity is specific to KT and it is hard to define monotonic relation between two word tokens in NLP. Also, we need to fix the response that to be inserted or deleted for imposing monotonic constraints, which does not make sense in NLP. The ablation studies on the monotonicity constraints in Section 3.2 shows that the direction of the monotonicity bias is important, which is also the unique nature of KT.
>
> Q2. Would consistency training leads to more improvements when the training data is limited?
>
> A2. Figure 3 shows that our method leads more improvement for smaller training data. We expect that consistency training alone is also more effective for smaller training data. We will provide supplementary experimental results for this in the appendix of the revision.
>
> Q3. How would the hyperparameter in insertion, deletion and replacements impact the performance?
>
> A3. The central hyperparameters for our approaches are augmentation probabilities and regularization loss weights. Our experience tells us that the performance improvements are much more sensitive to the loss weights, rather than the augmentation probabilities. We are going to add ablation results on the selection of these hyperparameters in the appendix of the revision.
>
> Q4. Would more advanced augmentation lead to better performance?
>
> A4. Absolutely. To the best of our knowledge, there weren’t any works that applied data augmentation strategies for knowledge tracing. The three augmentations we suggested are simple yet effective augmentations, but there are much more possibilities that we can exploit further, especially the augmentations for student learning interaction sequences that help to improve knowledge tracing or other educational tasks. More advanced augmentations and regularizations will be definitely worth investigating in the future, where our work will be an important guideline for them.
>
> [1] Xie et. al. Unsupervised Data Augmentation for consistency training, arXiv preprint.
> [2] Asai et. al. Logic-Guided Data Augmentation for Consistent Question Answering, arXiv preprint
>
> --------
> UPDATES: For Q2, we experimented with the DKT model on small datasets, only with replacement augmentation and consistency regularization loss, which also gives a significant improvement in AUCs. For example, on ASSISTmentsChall dataset with only 5% of train sets, the AUC is improved from 64.1% to 68.06% when we only use the consistency regularization, where we obtained AUC 69.09% with both consistency and monotonicity regularizations.

---

### Official Review · AnonReviewer4 · 2020-10-31
**This paper tackled the problem of knowledge tracing in education by proposing three data augmentation methods and demonstrated the effectiveness of the proposed methods on four widely-used datasets. However, the paper is limited in providing a clear connection between the proposed methods and previous studies, and did not provide an adequate description of relevant studies.**

**Rating:** 5
**Confidence:** 4

**Review:**

Knowledge tracing is a longstanding task in educational data mining and has been tackled by various studies. This paper proposed that three data augmentation methods (along with different types of regularization losses) can be applied to boost the performance of existing deep neural network models for knowledge tracing. Overall, the methods developed by this paper seem technically sound. In particular, the experiments are rather extensive, i.e., four widely-used datasets were employed in the experiments and different variants of the methods were investigated and compared. However, my biggest concern for this paper is its connection with previous studies and the design principles behind the proposed methods. To be specific, there are a few places that need to be further justified or a more clear explanation.

1. It would be good to provide a more detailed description of existing methods for knowledge tracing, e.g., what their limitations are and how the methods proposed can (theoretically) overcome their limitations?
2. In the Introduction section, it would be good to further justify "e.g., the recent GPT-3 model (Brown et al., 2020) has 175 billion parameters, they may easily overfit on small datasets and hurt model’s generalizability.". Any other evidence to show that overfitting is a common problem in existing deep neural network models for knowledge tracing? For existing works or the current study?
3. Also, it would be good to provide additional data analysis results to support the assumption behind the three data augmentation approaches? For example, in the existing datasets, to what extent can we observe that "a student is more likely to answer correctly (or incorrectly) if the student did the same more in the past"? In the experiments, to what extent such interaction sequences that were mistakenly modeled by previous studies can be modeled accurately by the newly-proposed methods?

---

> ### Author Response · Authors · 2020-11-12
> **Response to R4**
>
> Thank you for your valuable efforts and time spent on reading our paper. Our responses to all your questions and comments are provided below.
>
> Q1. It would be good to provide a more detailed description of existing methods for knowledge tracing, e.g., what their limitations are and how the methods proposed can (theoretically) overcome their limitations?
>
> A1. To the best of our knowledge, our work is the first approach in the literature that develops KT-specific data augmentation strategies. Namely, we study an unexplored problem and an apple-to-apple comparison with prior works is arguable.
>
> Domain-specific data augmentation has recently gained much attention as an effective way to impose domain-specific bias in the machine learning or computer vision community, e.g., consistency and contrastive learning as we mentioned in Section 1.1. However, they mostly focus on image datasets, and no such data augmentation was explored in prior works for KT. We believe that our work can be a strong guideline when other researchers will pursue to improve the generalization ability of KT models in the future.
>
> Q2. In the Introduction section, it would be good to further justify "e.g., the recent GPT-3 model (Brown et al., 2020) has 175 billion parameters, they may easily overfit on small datasets and hurt model’s generalizability.". Any other evidence to show that overfitting is a common problem in existing deep neural network models for knowledge tracing? For existing works or the current study?
>
> A2. Overfitting is a common issue for any neural networks in the regime of small datasets. Figure 3 indeed confirms that overfitting indeed occurs for KT models, as our method is more effective for smaller datasets in overall.
>
> Q3. Also, it would be good to provide additional data analysis results to support the assumption behind the three data augmentation approaches? For example, in the existing datasets, to what extent can we observe that "a student is more likely to answer correctly (or incorrectly) if the student did the same more in the past"? In the experiments, to what extent such interaction sequences that were mistakenly modeled by previous studies can be modeled accurately by the newly-proposed methods?
>
> A3. Thank you for your suggestion. We will add the data analysis results that actually show such characteristics of student interaction datasets we used in the revision.

---

### Author Response · Authors · 2020-11-12
**Our response before delivering the revision**

Dear reviewers,

We express our deepest gratitude for your constructive feedback and incisive comments on our manuscript.

We will carefully revise and enhance the manuscript with the additional experiments and discussions, which we hope to deliver soon.

Before that, we first plan to respond to questions and concerns you raised. We also appreciate your continued effort to provide further feedback until the very end of response/discussion phase. We will make sure to reflect all comments in the revision.

Thanks,
Authors.

---

### Author Response · Authors · 2020-11-20
**Summary of Revisions**

Dear reviewers,

Many thanks again for your constructive feedback to improve our manuscript. We have carefully incorporated your comments into this revision, as summarized in what follows:

* For R4, data analysis that shows the monotonicity nature of student interaction datasets, by observing the distributions of past interactions' correctness rates when the response correctness of the current interaction is fixed (Introduction, Appendix A.2)
* For R4, comparison of consistency loss for correctly and incorrectly predicted interactions, which supports the claim that smaller consistency loss actually improves prediction performances (Appendix A.3)
* For R3, grid-search results of augmentation strategies (Appendix A.4)
* For R2, another variation of replacement: skill-set-based replacement for the cases when questions can have multiple skills (ASSISTmentsChall and EdNet-KT1 datasets, Section 3.2)
* For R1, improved clarity (Figure 2, Table 1)
* For R2, additional references & fixing typos (Abstract, Introduction, Related works and Preliminaries, Figure 4)
* For R1, changes the order of figure (Figure 4)

These revisions are temporarily highlighted in "red" for your convenience.

If you have time, please check this revised manuscript and our previous response, and let us know if there are any other concerns to be clarified. We will be happy to respond to your further comments during the remainder of the author discussion period.

Best regards,

Authors

---

### Decision · Program_Chairs · 2021-01-07
**Final Decision**

**Decision:**

Reject

**Comment:**

The paper proposes new techniques for improving the generalization ability of deep learning models for Knowledge Tracing (KT). Instead of designing more sophisticated models, the paper investigates simple data augmentation techniques that can be applied to train existing models. In particular, three different augmentation strategies are proposed based on replacement, insertion, and deletion in the training data. These strategies are then applied with appropriate regularization loss ensuring consistency and monotonicity in the training process. Extensive experiments are performed using three popular neural models for KT and four publicly available datasets. Overall, the paper studies an interesting problem in an important application domain of online education. The results are promising and open up several exciting follow-up research directions to explore more complex data augmentation techniques for KT.

I want to thank the authors for actively engaging with the reviewers during the discussion phase and sharing their concerns about the quality of the reviews.  The reviewers generally appreciated the paper's ideas; however, there was quite a bit of spread in the reviewers' assessment of the paper (scores: 4, 5, 6, 7). In summary, this is a borderline paper, and unfortunately, the final decision is a rejection. The reviewers have provided detailed and constructive feedback for improving the paper. In particular, the authors should incorporate the reviewers' feedback to better position the work w.r.t. the existing literature on data augmentation and state of the art results, better motivate the data augmentation strategies in the context of educational applications possibly through additional data analysis, and add more ablation studies w.r.t. the hyperparameters associated with data augmentation. This is exciting and potentially impactful work, and we encourage the authors to incorporate the reviewers' feedback when preparing future revisions of the paper.